# *Public Wisdom Matters*! Discourse-Aware Hyperbolic Fourier Co-Attention for Social-Text Classification

**Karish Grover**
IIIT Delhi
India
karish19471@iiitd.ac.in

**S.M. Phaneendra Angara**
LinkedIn
India
sangara@linkedin.com

**Md. Shad Akhtar**
IIIT Delhi
India
shad.akhtar@iiitd.ac.in

**Tanmoy Chakraborty**
IIT Delhi
India
tanchak@ee.iitd.ac.in

## Abstract

Social media has become the fulcrum of all forms of communication. Classifying social texts such as fake news, rumour, sarcasm, etc. has gained significant attention. The surface-level signals expressed by a social-text itself may not be adequate for such tasks; therefore, recent methods attempted to incorporate other intrinsic signals such as user behavior and the underlying graph structure. Oftentimes, the 'public wisdom' expressed through the comments/replies to a social-text acts as a surrogate of crowd-sourced view and may provide us with complementary signals. State-of-the-art methods on social-text classification tend to ignore such a rich hierarchical signal. Here, we propose `Hyphen`, a discourse-aware hyperbolic spectral co-attention network. `Hyphen` is a fusion of hyperbolic graph representation learning with a novel Fourier co-attention mechanism in an attempt to *generalise* the social-text classification tasks by incorporating *public discourse*. We parse public discourse as an Abstract Meaning Representation (AMR) graph and use the powerful hyperbolic geometric representation to model graphs with hierarchical structure. Finally, we equip it with a novel Fourier co-attention mechanism to capture the correlation between the source post and public discourse. Extensive experiments on four different social-text classification tasks, namely detecting fake news, hate speech, rumour, and sarcasm, show that `Hyphen` generalises well, and achieves state-of-the-art results on ten benchmark datasets. We also employ a sentence-level fact-checked and annotated dataset to evaluate how `Hyphen` is capable of producing *explanations* as analogous evidence to the final prediction. Code is available at: https://github.com/LCS2-IIITD/Hyphen.

## 1 Introduction

Social media has become a significant source of communication and information sharing. Mining texts shared on social media (*aka* social-texts) are indispensable for multiple tasks – online offence detection, sarcasm identification, sentiment analysis, fake news detection, etc. Despite the proliferation of research in social computing, there is a gap in capturing the heterogeneous signals beyond the standalone source text processing. Predictive models incorporating signals such as user profiles [1, 2, 3, 4], underlying user interaction networks [5, 6, 7, 8, 9] and metadata information [10, 11, 12, 13], are far and few in between. These heterogeneous signals are challenging to obtain and may not always be available on different platforms (e.g., Reddit does not provide explicit user interaction network; YouTube does not release user activities publicly). On the other hand, comment

36th Conference on Neural Information Processing Systems (NeurIPS 2022).

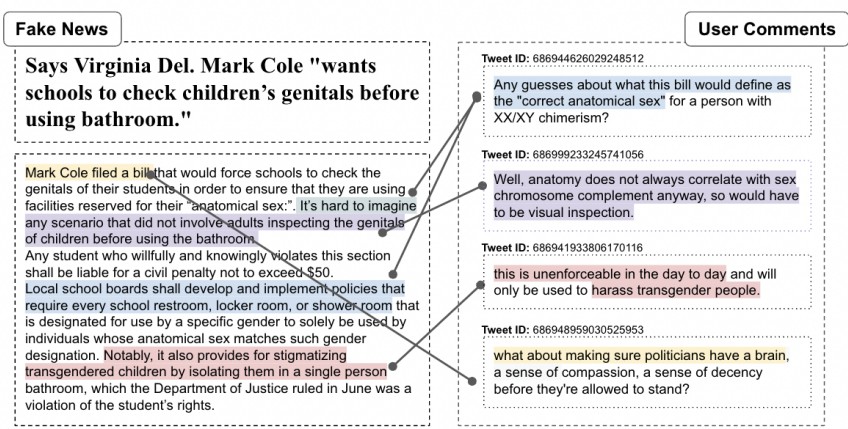

Figure 1: A motivating example (taken from our dataset) showing how user comments act as analogous evidence for a fake news article. The third comment hints towards a possible sense of harassment being brought out by the highlighted portion of the text (red) and that it is a possible fake news.

threads following a source post are an equally rich source of heterogeneous signals, which are easier to obtain and uniformly available across social media platforms and forums. We hypothesise that such public discourse carries complementary and rich latent signals (public wisdom, worldly knowledge, fact busting, opinions, emotions, etc.), which would otherwise be difficult to obtain from just standalone source-post analysis. Therefore, public discourse can be used in unison with the source posts to enhance social-text classification tasks. Figure 1 hints towards the motivation behind using public discourse as an implicit proxy for social-text classification.

In this work, we propose Hyphen, a discourse-aware hyperbolic spectral co-attention network that amalgamates the source post and its corresponding public discourse through a novel framework to perform generalised social-text classification. We parse individual comments on a source post as separate Abstract Meaning Representation (AMR) graphs [14], and merge them into one macro-AMR, representing mass perception and public wisdom about the source post. The AMR representation inherently abstracts away from syntactic representations, i.e., sentences which are similar in meaning are assigned similar AMRs, even if they are not identically worded [15]. The resultant macro-AMR graph represents the semantic information in a rich hierarchical structure [16, 17]. Hyphen aims to effectively utilise the hierarchical properties of the macro-AMR graph by using the hyperbolic manifold [18] for representation learning.

In order to fuse the source post with the public discourse, we propose a novel Fourier co-attention mechanism on the hyperbolic space. It computes pair-wise attention between user comments and the source post, thereby capturing the correlation between them. On a typical social media post, several users express their opinions, and there are several messages being conveyed by the source post itself, some of which are more relevant and/or common than the others. We use a novel discrete Fourier transform based [19] sublayer to filter the most-common user opinions expressed in the macro-AMR and most prominent messages being conveyed by the source post. Fourier transform is essentially a measurement of energy (i.e., strength of prevalence) of a particular frequency within a signal. We can extend this notion to quantify how dominant a particular frequency is within a signal. Building on this, we hypothesise that the time-domain signal isomorphically represents various user comments on the source post, and the Fourier transform over the comment representations yields the most-commonly occurring user *frequencies* (stance, opinions, interpretation, wisdom, etc.). Similarly, the Fourier transform over the sentence-level representations of the source post renders the *most intense* messages and facts being conveyed by it.

We perform extensive experiments with Hyphen on four social-text classification tasks – detecting fake news, hate speech, rumour, and sarcasm, on ten benchmark datasets. Hyphen achieves state-of-the-art results across all datasets when compared with a suite of generic and data-specific baselines. Further, to evaluate the efficacy of hyperbolic manifold and Fourier co-attention in Hyphen, we perform extensive ablation studies, which provide empirical justification behind the superiority of Hyphen. Finally, we show how Hyphen excels in producing explainability.

## 2 Related Work

**Generic social-text classification.** There have been some attempts to arrive at a general architecture for social-text classification. Bi-RNODE [20] proposes to use recurrent neural ordinary differential equations by considering the time of posting. CBS-L [21] considers transformation of document representation from the traditional $n$-gram feature space to a center-based similarity (CBS) space to solve the issue of co-variate shift. Pre-trained Transformer-based models like RoBERTa-base [22], BERTweet [23], ClinicalBioBERT [24], etc. also deliver benchmark results on generic social-text classification [25]. FNet [26] proves to be competent at modeling semantic relationships by replacing the self-attention layer in a Transformer encoder with a standard, non-parametric Fourier transform.

**Use of public discourse in social-text classification.** Multiple approaches have been proposed to use public discourse as an attribute for classifying the social media posts. TCNN-URG [27] utilises a CNN-based network to encode the content, and a variational autoencoder for modeling user comments in fake news detection. CSI [28] is a hybrid deep learning model that utilizes subtle clues from text, user responses, and the source post, while modeling the source post representation using an LSTM-based network. Zubiaga et al. [29] use public discourse for rumour stance detection using sequential classifiers. Lee et al. [30] propose sentence-level distributed representation for the source post guided by the conversational structure. CASCADE [31] and CUE-CNN [32] use stylometric and personality traits of users in unison with the discussion threads to learn contextual representations for sarcasm detection. dEFEND [33] and GCAN [9] propose to use co-attention over user comments and other social media attributes for detecting fake news and other social texts. The performance of most of these models deteriorate when extended to multiple tasks and fail to filter out the least relevant parts of their respective input modalities. Moreover, they operate on the Euclidean manifold, and therefore, overlook the representation strength of hyperbolic geometry in modeling hierarchical structures. `Hyphen` overcomes these limitations of the existing methods.

**Hyperbolic representation learning.** Hyperbolic representation learning has gained significant attention in tasks in which the data inherently exhibits a hierarchical structure. HGCN [34] and HAT [35] achieve state-of-the-art results in graph classification owing to their powerful representation ability to model graphs with hierarchical structure. Unlike these two, H2H-GCN [36] directly works on the hyperbolic manifold to keep global hyperbolic structure, instead of relying on the tangent space. Furthermore, the recent GIL model [37] captures more informative internal structural features with low dimensions while maintaining conformal invariance of both Euclidean and hyperbolic spaces. However, for social-text classification, none of the above approaches simultaneously consider the source- and discourse-guided representations. We build on this limitation and use public comments in unison with the source post to further contextualise and improve a social-text classifier.

## 3 Background

**Hyperbolic geometry.** A Riemannian manifold $(\mathcal{M}, g)$ of dimension $n$ is a real and smooth manifold equipped with an inner product on *tangent* space $g_{\boldsymbol{x}} : \mathcal{T}_{\boldsymbol{x}}\mathcal{M} \times \mathcal{T}_{\boldsymbol{x}}\mathcal{M} \to \mathbb{R}$ at each point $\boldsymbol{x} \in \mathcal{M}$, where the *tangent* space $\mathcal{T}_{\boldsymbol{x}}\mathcal{M}$ is an $n$-dimensional vector space and can be seen as a first-order local approximation of $\mathcal{M}$ around point $\boldsymbol{x}$. In particular, hyperbolic space $(\mathbb{H}^n_c, g^c)$, a constant negative curvature Riemannian manifold, is defined by the manifold $\mathbb{H}^n_c = \{\boldsymbol{x} \in \mathbb{R}^n : c\|\boldsymbol{x}\| < 1\}$ equipped with the following Riemannian metric: $g^c_{\boldsymbol{x}} = \lambda^2_{\boldsymbol{x}} g^E$, where $\lambda_{\boldsymbol{x}} = \frac{2}{1-c\|\boldsymbol{x}\|^2}$, and $g^E = \boldsymbol{I}_n$ is the Euclidean metric tensor. The connections between hyperbolic space and *tangent* space are established by the *exponential* map $\exp^c_{\boldsymbol{x}} : \mathcal{T}_{\boldsymbol{x}}\mathbb{H}^n_c \to \mathbb{H}^n_c$, and the *logarithmic* map $\log^c_{\boldsymbol{x}} : \mathbb{H}^n_c \to \mathcal{T}_{\boldsymbol{x}}\mathbb{H}^n_c$, as follows,

$$\exp^c_{\boldsymbol{x}}(\mathbf{v}) = \boldsymbol{x} \oplus_c \left(\tanh\left(\sqrt{c}\frac{\lambda^c_{\boldsymbol{x}}\|\mathbf{v}\|}{2}\right)\frac{\mathbf{v}}{\sqrt{c}\|\mathbf{v}\|}\right) \tag{1}$$

$$\log^c_{\boldsymbol{x}}(\boldsymbol{y}) = \frac{2}{\sqrt{c}\lambda^c_{\mathbf{v}}}\tanh^{-1}\left(\sqrt{c}\|-\mathbf{v}\oplus_c\mathbf{v}\|\right)\frac{-\mathbf{v}\oplus_c\mathbf{v}}{\|-\boldsymbol{x}\oplus_c\mathbf{v}\|} \tag{2}$$

where $\boldsymbol{x}, \boldsymbol{y} \in \mathbb{H}^n_c$, $\boldsymbol{v} \in \mathcal{T}_{\boldsymbol{x}}\mathbb{H}^n_c$, and $\oplus_c$ represents *Möbius addition* as follows,

$$\boldsymbol{x} \oplus_c \boldsymbol{y} = \frac{(1 + 2c\langle\boldsymbol{x}, \boldsymbol{y}\rangle + c\|\boldsymbol{y}\|^2)\boldsymbol{x} + (1 - c\|\boldsymbol{x}\|^2)\boldsymbol{y}}{1 + 2c\langle\boldsymbol{x}, \boldsymbol{y}\rangle + c^2\|\boldsymbol{x}\|^2\|\boldsymbol{y}\|^2} \tag{3}$$

Further, the generalization for multiplication in hyperbolic space can be defined by the *Möbius matrix-vector multiplication* between vector $\mathbf{x} \in \mathbb{H}_c^n \setminus \{\mathbf{0}\}$ and matrix $\mathbf{M} \in \mathbb{R}^{m \times n}$ as shown below,

$$\mathbf{M} \otimes_c \mathbf{x} = \frac{1}{\sqrt{c}} \tanh \left( \frac{\|\mathbf{Mx}\|}{\|\mathbf{x}\|} \tanh^{-1}(\sqrt{c}\|\mathbf{x}\|) \right) \frac{\mathbf{Mx}}{\|\mathbf{Mx}\|} \tag{4}$$

Hyperbolic space has been studied in differential geometry under five isometric models [18]. This work mostly confines to the Poincaré ball model. It is a compact representation of the hyperbolic space and has the principled generalizations of basic operations (e.g., addition, multiplication). We use $\mathcal{P}, \mathcal{E}$ in the superscript, to denote the Poincaré and Euclidean manifolds, respectively. We provide more insights into these models in Appendix A.1.

**Discrete Fourier Transform.** The Fourier transform decomposes a function into its constituent frequencies. Given a sequence $\{x_n\}$ with $n \in [0, N-1]$, the Discrete Fourier Transform (DFT) is defined as, $X_k = \sum_{n=0}^{N-1} x_n e^{-\frac{2\pi i}{N} nk}$, where $0 \leq k \leq N-1$. For each $k$, the DFT generates a new representation $X_k$ as a sum of the original input tokens $x_n$, with the *twiddle factors* [38, 19, 39].

## 4 Architecture of Hyphen

In this section, we lay out the structural details of Hyphen (see Figure 2 for the schematic diagram). We propose individual pipelines for learning representations of the source post and the user comments on the hyperbolic space. We then combine both the representations using a novel hyperbolic Fourier co-attention mechanism that helps in simultaneously attending to both the representations. Lastly, we pass it to a feed-forward network for the final classification. Without loss of generality, we denote the Poincaré ball model ($\mathcal{P}$) as $\mathbb{H}_c^n$ (hyperbolic space) throughout the paper.

### 4.1 Encoding public discourse

In this section, we discuss the pipeline for encoding the public discourse. We parse the user comments into an AMR (Abstract Meaning Representation) [14] graph. The individual comment-level AMR graphs are merged to form a macro-AMR (discussed below), representing the global public wisdom and latent *frequencies* in the discourse. Next, we learn representations of the macro-AMR using a HGCN (Hyperbolic Graph Convolutional Network) [40]. This yields a representation for public discourse containing rich latent signals.

**Macro-AMR graph creation**. Considering a social media post containing several user comments $C = [c_1, c_2, ..., c_m]$, we obtain an AMR (Abstract Meaning Representation) [14] graph for each user comment. We merge all the comment-level AMR graphs into one macro-AMR (post-level) while preserving the structural context of the subgraphs (comment-level). Figure 2(a) contains the schematic for an example macro-AMR graph. In particular, we adopt three strategies – (a) *Add a global dummy-node*: We add a dummy node and connect it to all the root nodes of the comment-level AMRs, and add a comment tag :COMMENT to the edges. The dummy node ensures that all the AMRs are connected, so that information can be exchanged during graph encoding. (b) *Concept merging*: Since we consider comments made on a particular post, these comments will essentially discuss the same topic. Therefore, multiple user comments can have identical mentions, resulting in repeated concept nodes in the comment-level AMRs. We identify such repeated concepts, and add an edge with label :SAME starting from *earlier* nodes to *later* nodes (here *later* and *earlier* refer to the temporal order of the ongoing conversation on a social media post). (c) *Inter-comment co-reference resolution*: A major challenge for conversational understanding is posed by pronouns, which occur quite frequently in such social media comments. We conduct co-reference resolution on the comment-level AMRs to identify co-reference clusters containing concept nodes that refer to the same entity. We add edges labeled with the label :COREF between them, starting from *earlier* nodes to *later* nodes in a co-reference cluster to indicate their relation. Such types of connections can further enhance cross-comments information exchange. This step results in a post-level AMR graph $\mathcal{G}_{amr} = [g_s^1, g_s^2, ..., g_s^m]$, representing relations between various subgraphs $\{g_s^i = (v_s^i, e_s^i) | 1 \leq i \leq m\}$ (different user comments will correspond to different subgraphs). The merged-AMR presents a global view of the public wisdom and interpretations.

**Hyperbolic graph encoder**. We adopt the Poincaré ball model of HGCN [40] to encode the post-level AMR graph and form user comment representations. Since different comments correspond to different

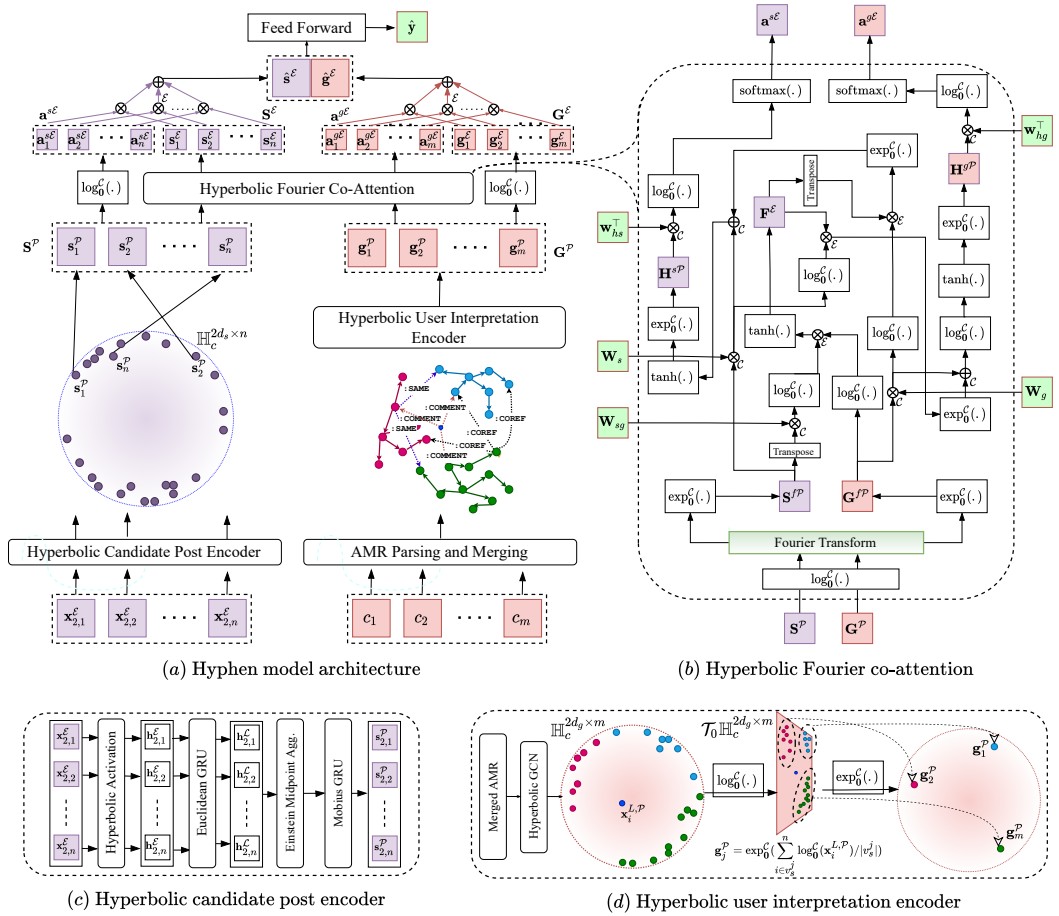

Figure 2: Dissecting the primary components of Hyphen. The overall model architecture shown in (a) contains two parallel pipelines to encode the candidate post and user comments; (c) encodes the candidate post's sentences using an attention-enhanced hyperbolic word encoder (Section 4.2), and (d) uses a hyperbolic GCN to encode the merged AMR containing latent user interpretations and form subgraph embeddings $\mathbf{g}_i^{\mathcal{P}}$ (Section 4.1). The final representations from (c) and (d), i.e., $\mathbf{S}^{\mathcal{P}}$ and $\mathbf{G}^{\mathcal{P}}$, are then passed to (b), which first transforms these through a Fourier sublayer and then computes the co-attention between user interpretation and the source post sentences in the hyperbolic space (Section 4.3).

subgraphs of the post-level AMR, we ultimately aggregate the node representations to form subgraph embeddings. Each subgraph embedding represents how individual users interpret the source post (their opinion). In this section, we summarize the graph encoder architecture. Given a post-level AMR graph $\mathcal{G}_{amr} = (\mathcal{V}, E)$ and the Euclidean input node features, denoted by $(\mathbf{x}^{0,\mathcal{E}}) \in \mathbb{R}^{d_g}$, where $d_g$ is the input embedding dimension for entities in the AMR graph, we first map the input from Euclidean to Hyperbolic space. Therefore, we interpret $\mathbf{x}^{\mathcal{E}}$ as a point in the tangent space $\mathcal{T}_{\mathbf{o}}\mathbb{H}_c^{d_g}$ and map it to $\mathbb{H}_c^{d_g}$ with $\mathbf{x}^{0,\mathcal{P}} = \exp_{\mathbf{0}}^c(\mathbf{x}^{0,\mathcal{E}})$. Our graph encoder then stacks multiple hyperbolic graph convolution layers to perform message passing (see Appendix A.2 for the background on HGCN). Finally, we aggregate the hyperbolic node embeddings $(\mathbf{x}_i^{L,\mathcal{P}})_{i\in\mathcal{V}}$ at the last layer to form subgraph (comments) embeddings as shown in Figure 2(d). We take the mean of the node embeddings for nodes present in a subgraph to yield the aggregated subgraph embedding: $\mathbf{g}_j^{\mathcal{P}} = \exp_{\mathbf{0}}^c(\sum_{i\in v_s^j}^n \log_{\mathbf{0}}^c(\mathbf{x}_i^{L,\mathcal{P}})/|v_s^j|)$. Here, $\mathbf{g}_j^{\mathcal{P}} \in \mathbb{H}_c^{2d_g}$, the operator $|.|$ represents the number of nodes present in subgraph $v_s^j$, and $L$ is the number of layers of HGCN. Therefore, the output for this encoder is $\mathbf{G}^{\mathcal{P}} = [\mathbf{g_1}^{\mathcal{P}}, \mathbf{g_2}^{\mathcal{P}}, ..., \mathbf{g_m}^{\mathcal{P}}]$, where $\mathbf{G}^{\mathcal{P}} \in \mathbb{H}^{2d_g \times m}$ is the matrix containing the learned representations for user interpretations (comments).

## 4.2 Hyperbolic Candidate Post Encoder

Inspired by [41], we propose to learn the source post content representations through a hierarchical attention network in the hyperbolic space. We know that not all sentences in a source post might contain relevant information. We thus employ a hierarchical attention-based network to capture the relative importance of various sentences. Consider the input embedding of the $t^{th}$ word appearing in the $i^{th}$ sentence as $\mathbf{x}_{it}$, in the candidate post. We utilise a hyperbolic word-level encoder (see Appendix A.3 for the background of Hyperbolic Hierarchical Attention Network (HHAN)) to learn $\mathbf{s}_i^{\mathcal{K}w}$, the representation of the $i^{th}$ sentence. Now, similar to the word-level encoder, we utilize *Möbius*-GRU units to encode each sentence in the source post. We capture the sentence-level context to learn the sentence representation $\mathbf{s}_i^{\mathcal{P}}$ from the sentence vector $\mathbf{s}_i^{\mathcal{P}w}$ obtained from the word-level encoder. Specifically, we use Poincaré ball model based *Möbius*-GRU to encode different sentences. We obtain outputs from the *Möbius*-GRU as $\mathbf{s}_i^{\mathcal{P}} = [\overrightarrow{GRU}_{mob}(\mathbf{s}_i^{\mathcal{P}w}), \overleftarrow{GRU}_{mob}(\mathbf{s}_i^{\mathcal{P}w})]$ as shown in Figure 2(e). Here, $\mathbf{s}_i^{\mathcal{P}}$ is the final context-aware representation for the $i^{th}$ sentence in the source post in the hyperbolic space (Poincaré ball model), i.e., $\mathbf{s}_i^{\mathcal{P}} \in \mathbb{H}_c^{2d_s}$, where $d_s$ is the input embedding dimension for the words $\mathbf{x}_{it}$ in the source post's sentences. This finally gives us $\mathbf{S}^{\mathcal{P}} = [\mathbf{s_1}^{\mathcal{P}}, \mathbf{s_2}^{\mathcal{P}}, ..., \mathbf{s_n}^{\mathcal{P}}]$, where $\mathbf{S}^{\mathcal{P}} \in \mathbb{H}_c^{2d_s \times n}$ is the matrix containing the learned candidate post representations.

## 4.3 Hyperbolic Fourier Co-Attention

We hypothesise that the evidence for various social-text classification tasks can be unveiled by investigating how different parts of the post are interpreted by different users, and how they correlate to different user opinions. Therefore, we develop a hyperbolic Fourier co-attention mechanism to model the mutual influence between the source social media post (i.e., $\mathbf{S}^{\mathcal{P}} = [\mathbf{s_1}^{\mathcal{P}}, \mathbf{s_2}^{\mathcal{P}}, ..., \mathbf{s_n}^{\mathcal{P}}]$) and user comments (interpretation) embeddings (i.e., $\mathbf{G}^{\mathcal{P}} = [\mathbf{g_1}^{\mathcal{P}}, \mathbf{g_2}^{\mathcal{P}}, ..., \mathbf{g_m}^{\mathcal{P}}]$, where $\mathbf{S}^{\mathcal{P}} \in \mathbb{H}_c^{2d_s \times n}$ and $\mathbf{G}^{\mathcal{P}} \in \mathbb{H}_c^{2d_g \times m}$). Co-Attention [42] enables the learning of pairwise attentions, i.e., learning to attend based on computing word-level affinity scores between two representations. Once we have the public discourse (Section 4.1) and the social media text (Section 4.2) embeddings in the hyperbolic space, the next step is a Fourier sublayer, which applies a 2D DFT to its (sequence length, hidden dimension) embedding input – one 1D DFT along the sequence dimension, $\mathcal{F}_{\text{seq}}$, and one 1D DFT along the hidden dimension, $\mathcal{F}_{\text{h}}$:[1]

$$\mathbf{S}^{f\mathcal{P}} = \exp_{\mathbf{0}}^c \left( \mathcal{F}_{\text{seq}} \left( \mathcal{F}_{\text{h}} \left( \log_{\mathbf{0}}^c (\mathbf{S}^{\mathcal{P}}) \right) \right) \right), \ \mathbf{G}^{f\mathcal{P}} = \exp_{\mathbf{0}}^c \left( \mathcal{F}_{\text{seq}} \left( \mathcal{F}_{\text{h}} \left( \log_{\mathbf{0}}^c (\mathbf{G}^{\mathcal{P}}) \right) \right) \right) \quad (5)$$

The intuition behind taking the Fourier transform over the user interpretation embeddings can be thought of as an attempt to capture the most commonly occurring *frequencies* (public wisdom, worldly knowledge, fact busting, opinions, emotions, etc.) in the public discourse. These *frequencies* signify how the source post is being received by most of the people. Further, the Fourier transform over the source post embeddings hints towards the most prominent messages conveyed by the source post. This is depicted in Figure 2(b). Next, we compute a proximity matrix $\mathbf{F}^{\mathcal{E}} \in \mathbb{R}^{m \times n}$. The affinity (proximity) matrix $\mathbf{F}^{\mathcal{E}}$ can be thought to transform the user-interpretation attention space to the candidate post attention space, and vice versa for its transpose $\mathbf{F}^{\mathcal{E}\top}$. It is computed as:

$$\mathbf{F}^{\mathcal{E}} = \tanh \left( \log_{\mathbf{0}}^c \left( \mathbf{S}^{f\mathcal{P}\top} \otimes_c \mathbf{W}_{sg} \right) \otimes_{\mathcal{E}} \log_{\mathbf{0}}^c \left( \mathbf{G}^{f\mathcal{P}} \right) \right) \quad (6)$$

where $\mathbf{W}_{sg} \in \mathbb{R}^{2d_s \times 2d_g}$ is a matrix of learnable parameters. The operator $\otimes_c$ is the *Möbius Multiplication* operator (Equation 4), and $\otimes_{\mathcal{E}}$ is the simple euclidean matrix multiplication. By treating the affinity matrix as a feature, we can learn to predict candidate post and user interpretation attention maps $\mathbf{H}^{s\mathcal{P}} \in \mathbb{H}_c^{k \times n}$ and $\mathbf{H}^{g\mathcal{P}} \in \mathbb{H}_c^{k \times m}$, given by

$$\mathbf{H}^{s\mathcal{P}} = \exp_{\mathbf{0}}^c(\tanh(\log_{\mathbf{0}}^c(\mathbf{W}_s \otimes_c \mathbf{S}^{f\mathcal{P}} \oplus_c \exp_{\mathbf{0}}^c(\log_{\mathbf{0}}^c(\mathbf{W}_g \otimes_c \mathbf{G}^{f\mathcal{P})) \otimes_{\mathcal{E}} \mathbf{F}^{\mathcal{E}\top}))))$$
$$\mathbf{H}^{g\mathcal{P}} = \exp_{\mathbf{0}}^c(\tanh(\log_{\mathbf{0}}^c(\mathbf{W}_g \otimes_c \mathbf{G}^{f\mathcal{P}} \oplus_c \exp_{\mathbf{0}}^c(\log_{\mathbf{0}}^c(\mathbf{W}_s \otimes_c \mathbf{S}^{f\mathcal{P})) \otimes_{\mathcal{E}} \mathbf{F}^{\mathcal{E}})))) \quad (7)$$

where $\mathbf{W}_s \in \mathbb{R}^{k \times 2d_s}, \mathbf{W}_g \in \mathbb{R}^{k \times 2d_g}$ are learnable parameters, $k$ is the latent-dimension used in computing co-attention and $\oplus_c$ is the *Möbius Addition* operator (Equation 3). We can then generate the attention weights of source words and interaction users through the Softmax function:

$$\mathbf{a}^{s\mathcal{E}} = \text{softmax}(\log_{\mathbf{0}}^c(\mathbf{w}_{hs}^{\top} \otimes_c \mathbf{H}^{s\mathcal{P}})), \quad \mathbf{a}^{g\mathcal{E}} = \text{softmax}(\log_{\mathbf{0}}^c(\mathbf{w}_{hg}^{\top} \otimes_c \mathbf{H}^{g\mathcal{P}})) \quad (8)$$

---

[1]The relative ordering of $\mathcal{F}_{\text{seq}}$ and $\mathcal{F}_{\text{h}}$ in Equation 5 is immaterial because the two 1D DFTs commute [26].

where $\mathbf{a}^{s\mathcal{E}} \in \mathbb{R}^{1 \times m}$ and $\mathbf{a}^{g\mathcal{E}} \in \mathbb{R}^{1 \times n}$ are the vectors of attention probabilities for each sentence in the source story and each user comment, respectively. $\mathbf{w}_{hs}, \mathbf{w}_{hg} \in \mathbb{R}^{1 \times k}$ are learnable weights. Eventually, we can generate the attention vectors of source sentences and user interpretation through weighted sum using the derived attention weights, given by

$$\hat{\mathbf{s}}^{\mathcal{E}} = \sum_{i=1}^{n} \mathbf{a}_i^{s\mathcal{E}} \mathbf{s}_i^{\mathcal{E}} \ , \quad \hat{\mathbf{g}}^{\mathcal{E}} = \sum_{j=1}^{m} \mathbf{a}_j^{g\mathcal{E}} \mathbf{g}_j^{\mathcal{E}} \tag{9}$$

where $\hat{\mathbf{s}}^{\mathcal{E}} \in \mathbb{R}^{1 \times 2d_s}$ and $\hat{\mathbf{g}}^{\mathcal{E}} \in \mathbb{R}^{1 \times 2d_g}$ are the learned co-attention feature vectors that depict how sentences in the source post are correlated to the user interpretations. Finally, we have a feed forward network which yields the final classification output as $\hat{\mathbf{y}} = FFN[\hat{\mathbf{s}}^{\mathcal{E}}, \hat{\mathbf{g}}^{\mathcal{E}}]$, where $[.]$ is the concatenation operator. Equipped with co-attention learning, our model is further capable of generating suitable explanations (Section 6) by looking into the co-attention weights between different *frequencies* of users and in the source post.

## 5  Experiments

**Datasets.** We evaluate the performance of Hyphen on four different social-text classification tasks across ten datasets (c.f. Table 1) – (i) fake news detection (Politifact [43], Gossipcop [43], AntiVax [44]), (ii) hate speech detection (HASOC [45]), (iii) rumour detection (Pheme [46], Twitter15 [47], Twitter16 [47], RumourEval [48]), and (iv) sarcasm detection (FigLang-Twitter[49], FigLang-Reddit [49]). We augmented the datasets with public comments/replies to suite our experimental setting (see the Appendix B for details on dataset preparation).

**Experimentation details.**
For both the hyperbolic encoders in Figures 2(c)-(d), we adopt the Poincaré model of the respective frameworks. Due to limited machine precision, it is possible that the $\exp_0^c(.)$ and $\log_0^c(.)$ maps might sometimes return points that are not exactly located on the manifold. To avoid this and to ensure that points remain on the manifold and tangent vectors remain on the right tangent space, we clamp the maximum norm to $1 - e^{-14}$. For optimization on

| Dataset | # source posts | Avg. comments (per post) | SOTA-1 | SOTA-2 | SOTA-3 |
|---|---|---|---|---|---|
| Politifact | 415 | 29 | *TCNN-URG [27] | HPA-BLSTM [50] | *CSI [28] |
| Gossipcop | 2813 | 20 | *TCNN-URG [27] | HPA-BLSTM [50] | *CSI [28] |
| Antivax | 3797 | 3 | *TCNN-URG [27] | HPA-BLSTM [50] | *CSI [28] |
| HASOC | 712 | 10 | CRNN | HPA-BLSTM | *CSI [28] |
| Twitter15 | 543 | 9 | BiGCN [51] | GCAN [9] | AARD [52] |
| Twitter16 | 362 | 27 | BiGCN [51] | GCAN [9] | AARD [52] |
| Pheme | 6425 | 17 | DDGCN [53] | *RumourGAN [54] | STS-NN [55] |
| Rumoureval† | 446 | 17 | CNN | DeClarE [56] | MTL-LSTM[57] |
| Figlang Twitter | 5000 | 4 | CNN + LSTM[58] | Ensemble {SVM, LSTM, CNN-LSTM, MLP}[59] | C-Net [60] |
| Figlang Reddit | 4400 | 3 | CNN + LSTM [58] | Ensemble {SVM, LSTM, CNN-LSTM, MLP} [59] | C-Net [60] |

Table 1: The statistics of the datasets and the chosen data-specific baselines for four social-text classification tasks. * denotes those baseline models which utilise public discourse. † denotes the dataset with three classes, and the remaining datasets have two levels.

the hyperbolic space, we use Riemannian Adam from Geoopt [61]. To find the optimal $k$ (latent dimension, see Equation 5) for hyperbolic co-attention, we run grid search over $k = 50, 80, 128, 256$, and finally use $k = 128$. For HGCN, we use two layers with curvatures $K_1 = K_2 = $ -1. We detail all other hyper-parameters in the Appendix B. We run all experiments for 100 epochs with early stopping patience of 10 epochs, on a NVIDIA RTX A6000 GPU.

**Curvature for our implementation**. Hyphen learns the hyperbolic representations for public discourse and source-post text simultaneously and applies a novel Fourier co-attention mechanism over the obtained embeddings. However, to be able to do so, we need to ensure that the curvatures of the hyperbolic manifolds (in our case *Poincaré ball* model) are same (or a product space of both manifolds). To ensure consistency across both the pipelines (public discourse encoder (Section 4.1) and source-post encoder (Section 4.2)), in Hyphen we take the constant negative curvature $c = -1$. As addressed in the limitations (See Section 7), another promising approach for Hyphen could be to consider the product space of both the manifolds before applying the co-attention mechanism.

**Baseline methods.** We compare Hyphen with two sets of baselines (c.f. Table 1) – **(i) Generic neural baselines:** We employ those models that are often used for social-text classification tasks and have

| Task | Dataset | | Data-specific baseline | | | Generic neural baseline | | | | Hyphen | |
|---|---|---|---|---|---|---|---|---|---|---|---|
| | | | SOTA-1 | SOTA-2 | SOTA-3 | HAN | dEFEND | BERT | RoBERTa | Eucli. | Hyper. |
| Fake News Detection | Politifact | Pre. | 0.712 | 0.894 | 0.847 | 0.852 | 0.902 | 0.911 | 0.924 | 0.951 | **0.972** |
| | | Rec. | 0.785 | 0.868 | 0.897 | 0.958 | 0.956 | 0.904 | 0.903 | 0.936 | **0.961** |
| | | F1 | 0.827 | 0.881 | 0.871 | 0.902 | 0.928 | 0.905 | 0.906 | 0.940 | **0.968** |
| | Gossipcop | Pre. | 0.715 | 0.684 | 0.732 | **0.818** | 0.729 | 0.764 | 0.771 | 0.786 | 0.791 |
| | | Rec. | 0.521 | 0.662 | 0.638 | 0.742 | 0.782 | 0.761 | 0.775 | 0.776 | **0.788** |
| | | F1 | 0.603 | 0.673 | 0.682 | 0.778 | 0.755 | 0.762 | 0.772 | 0.781 | **0.816** |
| | ANTiVax | Pre. | 0.829 | 0.865 | 0.901 | 0.806 | 0.935 | 0.943 | 0.948 | 0.941 | **0.951** |
| | | Rec. | 0.825 | 0.864 | 0.912 | 0.862 | 0.934 | 0.941 | **0.961** | 0.937 | 0.927 |
| | | F1 | 0.872 | 0.865 | 0.908 | 0.833 | 0.935 | 0.942 | 0.939 | 0.937 | **0.945** |
| Hate Speech Detection | HASOC | Pre. | 0.531 | 0.652 | 0.686 | 0.658 | 0.667 | 0.646 | 0.647 | 0.712 | **0.748** |
| | | Rec. | 0.529 | 0.697 | 0.699 | 0.681 | 0.672 | 0.651 | 0.661 | 0.703 | **0.718** |
| | | F1 | 0.591 | 0.634 | 0.698 | 0.614 | 0.657 | 0.641 | 0.648 | 0.702 | **0.713** |
| Rumour Detection | Pheme | Pre. | 0.785 | 0.816 | 0.846 | 0.821 | 0.841 | 0.861 | 0.852 | 0.854 | **0.877** |
| | | Rec. | 0.783 | 0.791 | 0.841 | 0.779 | 0.842 | 0.862 | 0.851 | 0.843 | **0.875** |
| | | F1 | 0.782 | 0.801 | 0.844 | 0.799 | 0.841 | 0.861 | 0.852 | 0.844 | **0.875** |
| | Twitter15 | Pre. | 0.866 | 0.824 | 0.928 | 0.929 | 0.851 | 0.899 | 0.913 | 0.943 | **0.961** |
| | | Rec. | 0.794 | 0.829 | 0.954 | 0.839 | 0.849 | 0.891 | 0.909 | 0.937 | **0.968** |
| | | F1 | 0.811 | 0.825 | 0.941 | 0.881 | 0.848 | 0.891 | 0.908 | 0.936 | **0.957** |
| | Twitter16 | Pre. | 0.871 | 0.759 | 0.901 | 0.941 | 0.892 | 0.921 | 0.895 | 0.944 | **0.946** |
| | | Rec. | 0.751 | 0.763 | **0.942** | 0.842 | 0.888 | 0.918 | 0.891 | 0.936 | 0.937 |
| | | F1 | 0.778 | 0.759 | 0.919 | 0.889 | 0.887 | 0.919 | 0.892 | 0.937 | **0.938** |
| | Rumour Eval | Pre. | 0.545 | 0.583 | 0.571 | 0.655 | 0.631 | 0.556 | 0.602 | **0.746** | 0.721 |
| | | Rec. | 0.676 | 0.777 | **0.888** | 0.444 | 0.555 | 0.533 | 0.602 | 0.686 | 0.718 |
| | | F1 | 0.598 | 0.667 | 0.695 | 0.518 | 0.573 | 0.533 | 0.595 | 0.697 | **0.712** |
| Sarcasm Detection | FigLang Twitter | Pre. | 0.701 | 0.741 | 0.751 | 0.734 | 0.758 | 0.797 | 0.822 | 0.811 | **0.823** |
| | | Rec. | 0.669 | 0.746 | 0.751 | 0.718 | 0.742 | 0.798 | 0.796 | 0.802 | **0.832** |
| | | F1 | 0.681 | 0.741 | 0.752 | 0.721 | 0.757 | 0.797 | 0.801 | 0.812 | **0.822** |
| | FigLang Reddit | Pre. | 0.595 | 0.672 | 0.679 | 0.671 | 0.639 | **0.723** | 0.691 | 0.707 | 0.712 |
| | | Rec. | 0.605 | 0.677 | 0.683 | 0.664 | 0.634 | 0.696 | 0.688 | 0.697 | **0.704** |
| | | F1 | 0.585 | 0.667 | 0.678 | 0.665 | 0.631 | 0.677 | 0.689 | 0.698 | **0.701** |

Table 2: Performance comparisons (Precision (Pre.), Recall (Rec.) and F1 score) of various baselines against `Hyphen-hyperbolic` (Hyper.) and `Hyphen-euclidean` (Eucli.). The best (*resp.* 2nd ranked) method is marked in bold (*resp.* underline). See Table 1 for other abbreviations.

been shown to perform comparatively. We consider different variations of the Transformer model and those who use social context as an auxiliary signal for social-text classification (dEFEND [33]). **(ii) Data-specific baselines**: We experimented with many data-specific and task-specific baselines and chose top three for every dataset based on the performance. Since top three models are data-specific, we call them with generic names – (a) SOTA-1, (b) SOTA-2, and (c) SOTA-3, respectively.

**Performance comparison.** Table 5 shows the performance comparison. The content-based pre-trained models, BERT and RoBERTa, outperform dEFEND which uses both the source content and user comments. We observe that dEFEND performs better than all the data-specific baselines because of the sophisticated use of co-attention. By incorporating public comments along with the social post, `Hyphen` shows significant[2] performance improvement over all the baselines. We observe that while the performance improvement over baselines is significant ($\sim 4\%$; $p < 0.005$) on datasets like Politifact, Gossipcop, and Twitter15, the performance improvement is not that significant ($p < 0.05$) on AntiVax and FigLang (Reddit). This is due to the fact that in the latter datasets, there are less number of comments available per the source posts (see Table 1). On Politifact and Gossipcop, `Hyphen-hyperbolic` has a performance gain of 3.9% and 3.8%, over the best baselines models, RoBERTa and dEFEND, respectively. Note that even when compared to the pre-trained Transformer architectures, `Hpyhen` shows decent improvement, while for the non-Transformer based baselines like HAN, there is a performance gain of 11.2% even on the AntiVax dataset. We explain the data-specific baselines, their modalities, and detailed analyses of their performance in the Appendix B.

**Ablation study.** We perform ablations with two variants of our model, namely `Hyphen-hyperbolic` and `Hyphen-euclidean`, in which Hyperbolic and Euclidean represent the underlying manifold.
■ **Effect of public wisdom.** When we remove user comments (`Hyphen w/o comments`: we consider only source post, get rid of the co-attention block for this analysis as we have just one modality, and keep the Fourier transform layer to capture the latent messages in the candidate post), the performance degrades. Table 3 shows that for Gossipcop and Politifact datasets, `Hyphen-hyperbolic` has a performance degradation of 7.23% and 7.4%, respectively. Due to the presence of less number of comments per post in the AntiVax dataset, the performance degradation is not that significant (i.e., 1.05%, $p < 0.1$). On some datasets like FigLang (Twitter), Pheme and Twitter15, `Hyphen-hyperbolic w/o comments` records a significant performance degradation ($p < 0.001$) of 8.47%,

---

[2]We also perform statistical significance $t$-test comparing `Hyphen` and the other baselines.

7.4% and 6.24%, respectively. Even `Hyphen-euclidean w/o comments` sees a fall in F1 score of 6.38%, 6.4% and 5.45% for Twitter15, Twitter16 and FigLang (Twitter), respectively. Since this is a content-only pipeline, in many cases, the model is outperformed by pre-trained Transformer models.

■ **Effect of hyperbolic space.** We evaluate `Hyphen`'s performance by replacing the hyperbolic manifold with Euclidean. We observe that in support of our initial hypothesis, `Hyphen-hyperbolic` outperforms `Hyphen-euclidean` (see Table 3). The former records a considerable gain of 3.55% and 2.85% F1 score on Gossipcop and Politifact datasets, respectively, over the latter. For the AntiVax dataset, a smaller increment of 0.83% can be attributed to the less number of user comments available in the dataset. Note that for the variant, `Hyphen-hyperbolic w/o comments`, there is a performance degradation as compared to `Hyphen-euclidean` on Gossipcop and Politifact. This is intuitive as the sole advantage of hyperbolic space lies in capturing the inherent hierarchy of the macro-AMR graphs. Therefore, in case of a content-only model, `Hyphen-euclidean` performs better. On Pheme and Twitter15, the former achieves a significant F1 score gain ($p < 0.005$) of 3.12% and 3.18% respectively. Due to less number of user comments in RumourEval, FigLang (Twitter) and Figlang (Reddit), the performance gain is less significant ($p < 0.05$), i.e., 1.44%, 1.01% and 0.47% respectively. It should be noted that this behaviour demonstrates the effectiveness of `Hyphen` in *early detection*. Even with less number of user comments available, `Hyphen` achieves performance boost over the baselines, and thus can be extremely effective in tasks like detecting fake news, where *early detection* is of great significance.

■ **Effect of Fourier transform layer.** Table 3 shows that including the Fourier transform layer to capture the most prominent user opinions about the source post and the most common (latent) messages conveyed by the source post, boosts the overall performance of `Hyphen`. There is an improvement of 5.6% F1 score on Gossipcop and 1.74% on Politifact due to the Fourier layer in `Hyphen-hyperbolic`. Because of the less number of comments per post in AntiVax, there is a smaller increment of 0.87% F1 score. Even for `Hyphen-euclidean`, there is an increase of 2.28% on Gossipcop and $\sim 1\%$ on Politifact. `Hyphen-hyperbolic` shows a significant improvement ($p < 0.001$) of 4.49%, 4.34%, and 4.13% F1 score on FigLang (Twitter), Pheme and Twitter15, respectively. For `Hyphen-euclidean`, there is an increase of 5.10% on FigLang (Twitter) and 3.53% on FigLang (Reddit). `Hyphen-euclidean` and `Hyphen-hyperbolic` record an average increase of 4.49% and 4.34% in F1 score, respectively, over all datasets. On applying co-attention over the outputs of Fourier transform layer, we are able to attend better to both the representations simultaneously, and thus the model's ability to capture the correlation between the two increases.

## 6 Explainability

Here, we demonstrate how `Hyphen` excels at providing explanations for social-text classification tasks. Using the hyperbolic co-attention weights $\mathbf{a}^{s\mathcal{E}}$ (Equation 8), we can provide an implicit rank list of sentences present in the source post in the order of their relevance to the final prediction. For instance, consider the scenario of fake news detection. A fake news is often created by manipulating selected parts of a true information. The generated rank list of sentences in this case would correspond to the sentences in the news article, which are possible misinformation. Furthermore, manual verification of all sentences in a news article is tedious, and therefore, a rank list based on the level of check-worthiness of sentences is convenient.

To evaluate the performance of `Hyphen` in generating explanations, we consider Politifact, and for each source post, we manually annotate sentences present in the source post based on their relevance to the final level (fake/real) (see Appendix C for dataset annotation details). The annotators also rank sentences of a source post in the order of their check-worthiness. We expect our model to produce a similar list of sentences for the source

| Model | Kendall's $\tau$ | Spearman's $\rho$ |
|---|---|---|
| dEFEND | $0.0231 \pm 0.053$ | $0.0189 \pm 0.012$ |
| Hyphen-euclidean | $0.4013 \pm 0.072$ | $0.4236 \pm 0.072$ |
| Hyphen-hyperbolic | $\mathbf{0.4983 \pm 0.055}$ | $\mathbf{0.5532 \pm 0.045}$ |

Table 4: Performance of `Hyphen` and dEFEND in providing explanations on Politifact. $\pm$ denotes std. dev. across 5 random runs.

post. We use dEFEND [33] as a baseline for comparing the rank correlations, and evaluate the rank list produced by `Hyphen` against this ground-truth *annotated rank list* using Kendall's $\tau$ and Spearman's $\rho$ rank correlation coefficients. dEFEND is the only model among the chosen baselines, which produces a similar rank link using attention weights in an attempt to provide explanations (See Appendix C for sample rank lists generated by dEFEND and `Hyphen`). Table 4 shows that the explanations produced by dEFEND have almost no correlation ($\tau = 0.0231, \rho = 0.0189$) to the annotated rank list. On the contrary, `Hyphen-hyperbolic` shows a high positive correlation

| Dataset | Model | Euclidean | | | Hyperbolic | | |
|---|---|---|---|---|---|---|---|
| | | Precision | Recall | F1 | Precision | Recall | F1 |
| **Politifact** | Hyphen | **0.9515** | **0.9364** | **0.9401** | **0.9722** | **0.9612** | **0.9686** |
| | Hyphen w/o comments | 0.9166 | 0.8802 | 0.8879 (↓ 4.22%) | 0.8461 | 0.9615 | 0.8963 (↓ 7.23%) |
| | Hyphen w/o Fourier | 0.9091 | 0.9523 | 0.9302 (↓ 0.99%) | 0.9341 | 0.9623 | 0.9512 (↓ 1.74%) |
| **Gossipcop** | Hyphen | **0.7862** | **0.7763** | **0.7812** | **0.7913** | **0.7884** | **0.8167** |
| | Hyphen w/o comments | 0.7557 | 0.7578 | 0.7551 (↓ 2.61%) | 0.7511 | 0.7734 | 0.7407 (↓ 7.60%) |
| | Hyphen w/o Fourier | 0.7751 | 0.7695 | 0.7584 (↓ 2.28%) | 0.7611 | 0.7812 | 0.7607 (↓ 5.60%) |
| **ANTiVax** | Hyphen | **0.9409** | **0.9375** | **0.9373** | **0.9511** | **0.9275** | **0.9456** |
| | Hyphen w/o comments | 0.9202 | 0.9187 | 0.9192 (↓ 1.81%) | 0.9417 | 0.9346 | 0.9351 (↓ 1.05%) |
| | Hyphen w/o Fourier | 0.9315 | 0.9281 | 0.9286 (↓ 0.87%) | 0.9365 | 0.9281 | 0.9369 (↓ 0.87%) |
| **HASOC** | Hyphen | **0.7121** | **0.7031** | **0.7031** | **0.7481** | **0.7187** | **0.7132** |
| | Hyphen w/o comments | 0.7122 | 0.6718 | 0.6693 (↓ 3.38%) | 0.6747 | 0.6718 | 0.6717 (↓ 4.15%) |
| | Hyphen w/o Fourier | 0.6909 | 0.6718 | 0.6762 (↓ 2.69%) | 0.7019 | 0.7031 | 0.6933 (↓ 1.99%) |
| **Pheme** | Hyphen | **0.8545** | **0.8437** | **0.8445** | **0.8771** | **0.8751** | **0.8757** |
| | Hyphen w/o comments | 0.8264 | 0.8142 | 0.8161 (↓ 2.84%) | 0.8121 | 0.7968 | 0.8017 (↓ 7.40%) |
| | Hyphen w/o Fourier | 0.8304 | 0.8203 | 0.8215 (↓ 2.30%) | 0.8411 | 0.8301 | 0.8323 (↓ 4.34%) |
| **Twitter15** | Hyphen | **0.9437** | **0.9375** | **0.9367** | **0.9703** | **0.9687** | **0.9685** |
| | Hyphen w/o comments | 0.8782 | 0.8751 | 0.8729 (↓ 6.38%) | 0.9078 | 0.9062 | 0.9061 (↓ 6.24%) |
| | Hyphen w/o Fourier | 0.9082 | 0.9062 | 0.9063 (↓ 3.04%) | 0.9444 | 0.9375 | 0.9272 (↓ 4.13%) |
| **Twitter16** | Hyphen | **0.9444** | **0.9363** | **0.9372** | **0.9464** | **0.9375** | **0.9382** |
| | Hyphen w/o comments | 0.9021 | 0.8751 | 0.8732 (↓ 6.40%) | 0.9196 | 0.9061 | 0.9042 (↓ 3.40%) |
| | Hyphen w/o Fourier | 0.9211 | 0.9071 | 0.9054 (↓ 3.18%) | 0.9067 | 0.9062 | 0.9155 (↓ 2.27%) |
| **RumourEval** | Hyphen | **0.7465** | **0.6862** | **0.6979** | **0.7219** | **0.7187** | **0.7123** |
| | Hyphen w/o comments | 0.6776 | 0.6364 | 0.6611 (↓ 3.68%) | 0.6941 | 0.6875 | 0.6898 (↓ 2.25%) |
| | Hyphen w/o Fourier | 0.7045 | 0.6875 | 0.6762 (↓ 2.17%) | 0.7433 | 0.6875 | 0.6743 (↓ 3.80%) |
| **FigLang_Twitter** | Hyphen | **0.8115** | **0.8025** | **0.8121** | **0.8235** | **0.8321** | **0.8222** |
| | Hyphen w/o comments | 0.7656 | 0.7583 | 0.7576 (↓ 5.45%) | 0.7555 | 0.7375 | 0.7375 (↓ 8.47%) |
| | Hyphen w/o Fourier | 0.7624 | 0.7617 | 0.7611 (↓ 5.10%) | 0.7779 | 0.7968 | 0.7773 (↓ 4.49%) |
| **FigLang_Reddit** | Hyphen | **0.7071** | **0.6979** | **0.6971** | **0.7107** | **0.7043** | **0.7018** |
| | Hyphen w/o comments | 0.6685 | 0.6511 | 0.6489 (↓ 4.82%) | 0.6743 | 0.6514 | 0.6513 (↓ 5.05%) |
| | Hyphen w/o Fourier | 0.6687 | 0.6642 | 0.6618 (↓ 3.53%) | 0.7091 | 0.6971 | 0.6961 (↓ 0.57%) |

Table 3: Ablation study showing the effect of public discourse, hyperbolic manifold, and Fourier transform layer on the performance of Hyphen for all four tasks. The decrease in performance of the ablation version of Hyphen w.r.t its original one is shown within parenthesis.

($\tau = 0.4983, \rho = 0.5532$). Hyphen-euclidean also shows comparable performance. The results present the efficacy of Hyphen in providing decent explanations for social-text classification.

**Model augmentation for explainability.** To provide explanations, we rule out the Fourier sub-layer from Hyphen. This is done because on taking the Fourier transform of source-post and comments' representations (Equation 5), we cannot assert an ordered mapping from the spectral domain to the sentence representations. Such an order is necessary for us to have a mapping between the co-attention weights and the source-post sentences they were derived from. Without such a mapping, Hyphen would not be able to generate a rank-list based on the sentences in the source-post.

## 7 Conclusion

Public wisdom on social media carries diverse latent signals which can be used in unison with the source post to enhance the social-text classification tasks. Our proposed Hyphen model uses a novel hyperbolic Fourier co-attention network to amalgamate both these information. Apart from the state-of-the-art performance in social-text classification, Hyphen shows the potential of generating suitable explanations to support the final prediction and works well in a *generalised* discourse-aware setting. In the future, mixed-curvature learning in product spaces [62] and hyperbolic-to-hyperbolic [36] representations could be employed to boost the learning capabilities of Hyphen.

**Limitations**. Hyphen resorts to using tangent spaces for computing Fourier co-attention which is inferior because tangent spaces are only a local approximation of the manifold. One may further incorporate other signals such as user interaction network and user credibility into the model.

## Acknowledgment

T. Chakraborty would like to acknowledge the support of the LinkedIn faculty research grant. We would like to acknowledge the support of the data annotators - Arnav Goel, Samridh Girdhar, Abhijay Singh, and Siddharth Rajput, for their help in annotating the Politifact dataset.

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
