# OpenReview forum: "Public Wisdom Matters! Discourse-Aware Hyperbolic Fourier Co-Attention for Social Text Classification"
_NeurIPS.cc/2022/Conference — NeurIPS 2022 Accept_

### Official Review · Reviewer_jzcw · 2022-07-08

**Rating:** 7
**Confidence:** 3
**Soundness:** 4 excellent
**Presentation:** 3 good
**Contribution:** 3 good

**Summary:**

Social-text classification is challenged by the brevity and lack of context in individual posts. Previous work has tried to augment predictive models with external (to the post) sources of information such as signals derived from users or their networks. However, such signals may not always be readily available. In contrast, public discourse is usually available in the form of comments or replies to a given post. This paper, proposes Hyphen, a novel model that exploits public discourse to improve social-text classification. The model consists of three main modules: candidate post encoder, public discourse encoder and co-attention between the representations induced by the encoders.

To encode the pubic discourse, individual comments are first represented as AMR graphs which are then merged into a single macro-AMR graph. The macro-AMR is then passed into a Hyperbolic Convolutional Neural Network to induce embedding representations.
The candidate post is encoded through a Hyperbolic Hierarchical Attention Network — individual word representations are transformed into sentence representations, using bi-directional Mobius-GRU layers with attention. Similarly, the final document representation is obtained  with a higher-level Mobius-GRU layer with attention that processes the sentence representations.
Both induced representations are then combined via an Hyperbolic Fourier Co-attention mechanism that produces affinity scores between representations. The representations are first transformed via a 2D Discrete Fourier Transform to extract the most salient (high “frequency”) features. Then an affinity matrix is learned such that it can produce attention weights between source words and user interactions, which are used to obtain the final weight aggregated representations for the source post and the public discourse signals. These attention weights can also be used to interpret the predictions: by ranking candidate sentences wrt to attention scores we can get a sense of which parts of the input most contribute to the predictions.

The evaluation performed on a variety of datasets and tasks such as detection of fake news, rumors, hate-speech and sarcasm, shows that the proposed model is effective.

**Questions:**

N/A

**Limitations:**

Yes, for the most part. However, given that this work entails collecting and analyzing personal data from social media, a discussion on potential negative societal impact is missing.

**Strengths And Weaknesses:**

This paper addresses an important problem in social-text classification: how to augment predictive models with external information? The proposed approach seems to be effective at leveraging public discourse associated to a target post by exploiting the hierarchical nature of public discourse (in social-texts). Another benefit of this model is the Fourier co-attention module that allows models to focus on the most important parts of the input. Furthermore, this mechanism can afford some degree of interpretability by inspecting the attention weights attributed to each sentence on a given input. However, claims of model interpretability based on attention must be taken with a grain of salt as previous work has shown that “Attention is Not Explanation” (Jain and Wallace, 2019).

Overall the paper is well written and organized, and the main claims of the paper are supported by experimental evidence. However, the notation is a bit convoluted and thus hard to follow with the various superscripts and subscripts. Perhaps the authors can polish and simplify the notation on the final draft. There are also broken references (e.g. line 192 refers to Figure (e) which does not exist; line 272 refers a Table 5 that does not exist; line 261 discusses optimizing k and points to Eq. 5 but this equation has no k).

---

> ### Author Response · Authors · 2022-08-01
> **Response to Reviewer jzcw: Explainability**
>
> We sincerely thank the reviewer for the valuable assessment and constructive feedback. We hereby address the raised questions and suggestions.
>
> > However, claims of model interpretability based on attention must be taken with a grain of salt as previous work has shown that “Attention is Not Explanation” (Jain and Wallace, 2019).
> >
> - We agree that vanilla attention might not be "explainability" in its truest sense. However, `Hyphen` uses a more sophisticated Hyperbolic co-attention, which empirically shows higher interpretability. This is due to its ability to filter out the public wisdom and to amalgamate the discourse with source posts for final prediction.
> - To evaluate the performance of `Hyphen` in generating explanations in a real-world scenario, we consider the Politifact dataset. For each source post, we manually annotate sentences present in the source post based on their relevance to the final level (fake/real). `dEFEND` achieves almost 0 correlation with the ground truth annotation in terms of explainability. `Hyphen`, on the other hand, shows a high correlation with the annotated rank-list, and thus empirically shows better interpretability (**Table 4, Section 6**).
> - Further, we would also like to bring up the previous work, "Attention is not not Explanation" *(Wiegreffe and Pinter, 2019)*, which was written in response to "Attention is Not Explanation" *(Jain and Wallace, 2019)*. They argue that whether or not attention is an explanation depends on one's definition of explanation and that testing it needs to consider several elements of the model. Our Explainability evaluation (**Section 6**) proves this hypothesis with sufficient improvement over "naive" co-attention.
>
> > Perhaps the authors can polish and simplify the notation on the final draft. There are also broken references (e.g. line 192 refers to Figure (e) which does not exist.
>
> We would like to thank the reviewer for bringing this up. We will surely poilsh and simplify the notation in the final draft, and add a notation table, if needed.

---

> ### Author Response · Authors · 2022-08-01
> **Response to Reviewer jzcw: Potential negative societal impacts.**
>
> > However, given that this work entails collecting and analyzing personal data from social media, a discussion on potential negative societal impact is missing.
>
> We hereby discuss some potential negative societal impacts of this work, mitigation strategies, and how the existing `Hyphen` framework tackles them:
>
> **Malicious or unintended uses.** Our work does not involve a generative framework and hence cannot be used *directly* to generate DeepFakes. However, similar to other classification models like `dEFEND`, when combined with a generative model like a variational autoencoder, it could be misused to generate fake user comments. This could be mitigated with the help of our Fourier co-attention block. `Hyphen` adopts a Fourier transform-based network, which converts the comments' space to the *frequencies* domain. Thus, it cannot be used directly for generating fake user comments on social media posts because the user comment representations are now in a different semantic space. To do this, one would have to modify the source code and get rid of the Fourier block.
>
> **Environmental impact.** `Hyphen` architecture could be used to train large-scale models for various social-text classification tasks. However, owing to the numerical instabilities, the Riemannian optimization for Hyperbolic geometry is highly complex (e.g., Due to limited machine precision, the exponential and logarithmic maps might sometimes return points that are not exactly located on the manifold. To avoid this and to ensure that points remain on the manifold and tangent vectors remain on the right tangent space, we clamp the maximum norm). Thus, even though it is possible to train `Hyphen` on a large scale, it would not come without complications.
>
> **Fairness considerations.** The intuition behind `Hyphen` lies in finding the most prominent *frequencies* from the public discourse. However, often, the fake news (or hate speech) spreaders bias the comment section of a social media post to lure others into believing them. This could involve a fake news author creating fake accounts to comment in favor of his agenda and asking his connections to do the same. If the comment section is not credible, finding the majority voice of the people might bias the system in favor of the majority (which is a "pseudo-majority" as malicious users created it to fool the detection bots). To mitigate this, an additional module that estimates the credibility of user comments (and users) could assist the model in making fair decisions.
>
> **Privacy considerations.** Our work operates within the bounds of privacy considerations. We utilize the public discourse from social media to act as a surrogate for user opinions and worldly wisdom. However, we do not consider the user profile features (such as user background) as these might violate some privacy concerns and lead to ethical problems like "user profiling" [1]. Instead, we consider the user interpretation (opinions and wisdom) just w.r.t. the social-media post in hand.
>
> We will add this discussion on the potential negative societal impacts in the final draft.
>
> [1] Mann, Monique, and Tobias Matzner. "Challenging algorithmic profiling: The limits of data protection and anti-discrimination in responding to emergent discrimination." Big Data & Society 6.2 (2019): 2053951719895805.

---

> ### Author Response · Authors · 2022-08-08
> **Following up rebuttal response**
>
> We sincerely thank the reviewer for the valuable assessment and constructive feedback. We have tried to address the raised questions and suggestions in our rebuttal. We have elaborated on the ability of Hyphen to generate suitable explanations and have included a discussion on the potential negative societal impacts. Could you kindly let us know your feedback on the same?

---

### Official Review · Reviewer_Lrhk · 2022-07-10

**Rating:** 8
**Confidence:** 3
**Soundness:** 3 good
**Presentation:** 3 good
**Contribution:** 3 good

**Summary:**

This paper addresses the problem of social media classification. In detail, it targets fake news, hate speech, rumour, and sarcasm detection on ten datasets. The approach combines hyperbolic graph representation learning with Fourier co-attention. In detail, the authors encode the candidate with an attention-enhanced hyperbolic word encoder and use abstract meaning representations to learn hyperbolic graph representations of posts aligned with a candidate. During the evaluation, they compare their approach with generic neural baselines (i.e. Bert) and data-specific baselines - like CNN-based approaches. They found statistically significant improvements in the majority of the tasks. The final ablation study examinates the effect of incorporating social media posts and using the Fourier co-attention mechanism,


**Questions:**

- How was a candidate and its post encoded?
- Does table 2 shows the average results over multiple runs using different seeds?


**Limitations:**

No limitations

**Strengths And Weaknesses:**

Strengths:
- The proposed method shows constant improvement over different tasks, especially when the number of posts is large. Moreover, it shows that it can gain even from an extensive collection of posts.
- Different aspects of this method are examined in detail during the ablation study and show their importance.
- Apart from the pure performance evaluation, the authors include a concluding evaluation regarding the interpretability of single sentence importance from the candidates based on their approach. They found a clear improvement in the baseline.

Weaknesses:
- The authors compared their approach to pre-trained language models - like Bert - and found overall tasks a constant performance improvement. From the paper, it is unclear how social media posts were presented to these models. In the case of concatenating the candidate and its posts, parts of these posts could be ignored by the models. Because they have a fixed token limitation and most datasets have many relevant social media posts.

---

> ### Author Response · Authors · 2022-08-01
> **Response to Reviewer Lrhk**
>
>
> We sincerely thank the reviewer for the valuable assessment and constructive feedback. We hereby address the raised questions and suggestions.
>
> > The authors compared their approach to pre-trained language models - like Bert - and found overall tasks a constant performance improvement. From the paper, it is unclear how social media posts were presented to these models. In the case of concatenating the candidate and its posts, parts of these posts could be ignored by the models. Because they have a fixed token limitation and most datasets have many relevant social media posts.
>
> We sincerely apologize for this confusion. The social media post (text) was directly fed into pre-trained models like BERT, and these models were fine-tuned for the respective social-text classification tasks. We would also like to stress that:
> - We consider only the source post as an input modality (and not the corresponding user comments) while fine-tuning; thus, there is no notion of "concatenation" here. We do not consider the social context (user comments) because these models initially have no sophisticated degrees of freedom for multiple modalities (except vanilla operations like concatenation).
> - For a fair comparison of `Hyphen` with baselines that consider both the source post and the user comments, we have other task-specific baselines and a general neural baseline, `dEFEND`. We have mentioned the experimental setup and input modalities of various data-specific baselines in the **Appendix B**: *Data-specific baselines*.
> - Moreover, the datasets used have an average token length of < 43 (because most of the datasets consist of tweets and Reddit posts). In datasets like Politifact and Gossipcop, where the post length is larger, it is still < 512 (max. length in BERT).
> - Furthermore, if the source post length was more than the maximum token length (as brought out in the question), this could be considered a limitation of BERT-like pre-trained models for any text classification task. `Hyphen` does not encounter such maximum tokens limitations owing to a hierarchical attention-based hyperbolic source post encoder and an AMR-based representation for the user comments (**Sections 4.1 and 4.2**).
> > How was a candidate and its post encoded?
>
> To encode the candidate post and the corresponding user comments for `Hyphen`, we adopt the following procedure:
>  - We adopt the Poincaré ball model of Hyperbolic GCN to encode the post-level AMR graph and form user comment representations (**Section 4.1**, *Figure 2(d)*).
>  - Further, we propose to learn the source post content representations through a hierarchical attention network in the hyperbolic space (**Section 4.2**, *Figure 2*(c)).
>  - We have also elaborated the architectural details for these in **Appendix A.2** and **Appendix A.3**, respectively.
>
>
> > Does table 2 shows the average results over multiple runs using different seeds?
>
> Yes, Table 2 shows average results for five runs using random seeds. We sincerely apologize for not mentioning this clearly in the paper; however, in the final revision, we shall explicitly mention this. Further, in Table 4, we have even mentioned the standard deviation for 5 random runs (*Hyphen explainability evaluation*).

---

> > ### Comment · Reviewer_Lrhk · 2022-08-09
> > **Updates**
> >
> > Thanks a lot to the authors for their detailed statements! That confirms my scoring.

---

> ### Author Response · Authors · 2022-08-08
> **Following up rebuttal response**
>
> We sincerely thank the reviewer for the valuable assessment and constructive feedback. We have tried to address the raised questions and suggestions in our rebuttal. Could you kindly let us know your feedback on the same?

---

### Official Review · Reviewer_NXDS · 2022-07-10

**Rating:** 7
**Confidence:** 3
**Soundness:** 3 good
**Presentation:** 4 excellent
**Contribution:** 2 fair

**Summary:**

This paper proposes to apply hyperbolic fourier co-attention mechanism to aggregate public wisdom in social text classification. In social text classification tasks, e.g. fake news detection and hate speech detection, previous works have proved that the comments or reactions from other users on the internet could contribute a lot in boosting the performance of text classification models. This work proposes to use a co-attention mechanism to aggregate the public wisdom within the Abstract Meaning Representation (AMR) graph constructed from comments and reactions. More specifically, they replace the operations like addition and mutiplication in original attention and Fourier transforms with the corresponding definitions in hyperbolic geometry. The experiment results show that their proposed works outperform both task specific SOTA and generic SOTA nerual models.

**Questions:**

Based on my understanding, the reason why the Hyperbolic geomerty works is that it can capture hierachy structure more effectively. Would the authors like to do some visualization to the learnt embedding to verify that the model does learn a good embedding?

**Limitations:**

See the weakness. Also, I think that the idea of aggregating the public wisdom in social text classification is well known. The main contribution of this work is in designing neural architectures to capture such wisdom better.

**Strengths And Weaknesses:**

Strengths:
1. The performance of the proposed model is impressive.
2. The idea to capture the hierarchy structure in the AMR graph is convincing.
3. I appreciate that the authors did so many experiments for the evaluation.

Weakness:
My main concern is about the comparison of task specific SOTA. The issue is that the data specific SOTAs are usually pre-BERT models. As a result, their backbones for capturing linguistic cues are generally weak. For example, the CSI model simply computes the mean of all word embeddings of the text for the detection. From the table 2, we can also see that those generic neural models almost defeat all those data specific SOTAs. As a result, it is hard to say that the proposed Fourier Hyperbolic Co-Attention mechanism can bring better boosts compared to the technologies applied in the data specific SOTAs. To address my concern, I expect the authors could either claim that those data specific SOTAs can not use BERT or provide some results of the data specific SOTAs with BERT backbone (not necessary on all SOTA or all datasets, since the time for rebuttal will be limited. Only several common SOTAs, like CSI, on several datasets would be helpful.)

---

> ### Author Response · Authors · 2022-08-01
> **Response to Reviewer NXDS: Data-specific SOTAs with BERT backbone.**
>
> We sincerely thank the reviewer for the valuable assessment and constructive feedback. We hereby address the raised questions and suggestions.
>
> > To address my concern, I expect the authors could either claim that those data specific SOTAs can not use BERT or provide some results of the data specific SOTAs with BERT backbone.
>
> Following the reviewer's valuable suggestion of augmenting the data-specific baselines with a BERT backbone for a fair comparison, we have made the following modifications to the baselines. Some baselines like `AARD` and `C-Net` already use BERT. For several other baselines, we carry out the modifications listed in the table below:
>
> | **Baseline** |                                  **Modification**                                 |
> |:------------|:---------------------------------------------------------------------------------|
> | TCNN-URG     | Replace the two-level CNN with BERT for encoding news articles.                   |
> | HPA-BLSTM    | Replace the hierarchical attention network with BERT encoder + Attention.         |
> | CSI          | Replace Doc2Vec with BERT for textual feature representation in news.             |
> | CRNN         | Add BERT embeddings instead of the Glove/FastText embedding layer.                |
> | AARD*        | Already uses BERT to obtain the representation of text contents (initialization). |
> | GCAN         | Replace Bi-GRU with BERT for encoding source tweets.                          |
> | RumourGAN    | Replace the GRU-RNN encoder with BERT for encoding text.                          |
> | STS-NN       | Add BERT embeddings instead of the word2vec embedding layer.                      |
> | MTL-LSTM     | Add BERT embeddings instead of the word2vec/Glove embedding layer.                |
> | CNN          | Add BERT embeddings instead of the Glove/FastText embedding layer.                |
> | C-Net*       | Already uses BERT to obtain the representation of text.                           |
> | DeClarE      | Replace Bi-GRU with BERT for text aggregation.                                |
> | CNN + LSTM   | Add BERT embeddings instead of the Glove/FastText embedding layer.                |
>
> The table below shows the F1 score for the above experiments, accross all datasets. The results are averaged over 5 random runs:
>
> | Dataset | Politifact | Gossipcop | ANTiVax | HASOC | Pheme | Twitter15 | Twitter16 | RumourEval | FigLang Twitter | FigLang Reddit |
> |:-------:|:----------:|:---------:|:-------:|:-----:|:-----:|:---------:|:---------:|:----------:|:---------------:|:--------------:|
> |  SOTA-1 |      0.841 |     0.628 |   0.881 | 0.607 |     - |         - |         - |      0.632 |           0.691 |          0.592 |
> |  SOTA-2 |      0.865 |     0.667 |   0.844 | 0.612 | 0.822 |     0.856 |     0.791 |      0.651 |           0.752 |          0.681 |
> |  SOTA-3 |      0.891 |     0.703 |   0.912 | 0.699 | 0.851 |         * |         * |      0.697 |               * |              * |
>
> (* denotes that the baseline model already uses a BERT backbone. - denotes that the baseline cannot use BERT.)
>
> The results show that substituting the models with a `BERT` backbone boosts the model performance in almost all cases. The F1 increment is around 2-3 % in most cases. However, it is also observed that in some baselines like `HPA-BLSTM` and `DeClaRe`, adding a BERT backbone decreases the model performance. This probably happens because we substitute a hierarchical attention-type network with BERT in both cases, and the former is extremely good at learning the word-level and sentence-level representations. We will add these results in the final draft.

---

> > ### Comment · Reviewer_NXDS · 2022-08-08
> > **Updates**
> >
> > Thank you for your efforts and responses! Your response answered my questions. I will raise my score.

---

> ### Author Response · Authors · 2022-08-01
> **Response to Reviewer NXDS: Visualization of Hyperbolic embeddings.**
>
>
> > Would the authors like to do some visualization to the learnt embedding to verify that the model does learn a good embedding?
>
> We thank the reviewer for the valuable suggestion to do some visualization of the learned embeddings. However, there are a few concerns about why visualization of embeddings to show the effectiveness of Hyperbolic Fourier co-attention may not be a feasible option in this case:
>
> ### No explicit embedding to visualize.
> - In `Hyphen`, the Hyperbolic Fourier transform block projects the sentence and user comments' embeddings in the hyperbolic space, in the "frequency" domain. These embeddings are then amalgamated together using the hyperbolic co-attention module. The final hyperbolic attention maps $\mathbf{H}^{s\mathcal{P}} \in \mathbb{H}_c^{k \times n}$ and $\mathbf{H}^{g\mathcal{P}} \in \mathbb{H}_c^{k \times m}$. It is difficult for a reader to understand the semantics of these attention maps in the form of visualization, as they do not essentially "represent" any input modality but are an intermediate bi-product of `Hyphen`.
> - One possible visualization could be that of the hyperbolic macro-AMR graph, which could show how well the Hyperbolic GCN is learning the graph embeddings. However, this visualization would not explicitly show the effectiveness of `Hyphen`, as these are, in fact, "fed into" the Hyperbolic Fourier co-attention block. This would show the effectiveness of GCN and not `Hyphen` as a whole. If the reviewer feels such a visualization could be beneficial, we would be happy to add such visualization in the final draft. However, such a visualization could face the second challenge discussed below.
>
> ### Dimentionality reduction.
> - As we know that visualizing embeddings of higher dimensions in any embedding space is not possible without projecting them to a lower dimensional 2D (or 3D) space (say using t-SNE or PCA algorithms). However, regarding Riemannian geometry, these algorithms' Euclidean variants cannot be directly applicable to the Hyperbolic embeddings.
> - There could be a possible way to visualize these higher-dimensional hyperbolic embeddings: Project these embeddings onto the tangential plane, apply t-SNE for dimensionality reduction, and visualize the resultant Euclidean embeddings. However, there is an inherent problem with this methodology. Firstly, projection onto the tangential place is but an approximation, and secondly, dimensionality reduction after this approximation would destroy the hyperbolic geometrical properties of these embeddings. As a result, the latent hierarchy would not survive to that extent, and these embeddings would look indistinguishable from the Euclidean embeddings to the naked eye. Some previous works like [1] show visualizations for Poincare ball embeddings; however, they were trained on 2-dimensional $\mathbb{H}_c^2$, and hence no dimensionality reduction was required for the visualization.
>
> [1] Nickel, Maximillian, and Douwe Kiela. "Poincaré embeddings for learning hierarchical representations." Advances in neural information processing systems 30 (2017).

---

### Official Review · Reviewer_niN4 · 2022-07-11

**Rating:** 6
**Confidence:** 4
**Soundness:** 3 good
**Presentation:** 3 good
**Contribution:** 3 good

**Summary:**

This paper proposes to leverage the public wisdom expressed in social media comments and replies to augment social text classification. To this end, this work proposes Hyphen, which consists of hyperbolic graph representation learning and a Fourier co-attention mechanism. Extensive experiments on 10 datasets demonstrate the effectiveness of Hyphen.

**Questions:**

1. I appreciate that the authors leverage 10 datasets for evaluation. Many of these datasets are well-explored in previous literature, and they often use different dataset splits. Maybe I missed it, but the authors did not seem to provide the train/validation/test split information of these 10 datasets. It would be helpful for readers to know if the authors followed the splits of any previous work or created splits in these experiments.

2. Did the authors implement "SOTA-1", "SOTA-2", and "SOTA-3" in Table 2 or directly quoted the reported results in these papers? It does not seem clear to me what the answer is. If the results are directly quoted, did the authors make sure that the train/validation/test split is identical?

3. The idea of using social discourse such as user comments or interactions on social media to augment social text classification is not new, for example, dEFEND. I hope the authors could better justify how introducing the maths of Hyperbolic Fourier Co-Attention actually help Hyphen achieve this goal.

4. While BERT and RoBERTa are indeed powerful models, the state-of-the-art of NLP has progressed from there and many improved LMs have emerged. In the spirit of [1], I suggest the authors check out newer LMs [2,3] as sota baselines against Hyphen.

5. I appreciate the authors' efforts in annotating the PolitiFact dataset for experiments in Section 6. Did the authors use crowdsourcing for these annotations? If so, please answer the questions in Checklist 5 accordingly.

[1] Bowman, Samuel. "The Dangers of Underclaiming: Reasons for Caution When Reporting How NLP Systems Fail." Proceedings of the 60th Annual Meeting of the Association for Computational Linguistics (Volume 1: Long Papers). 2022.

[2] Clark, Kevin, et al. "Electra: Pre-training text encoders as discriminators rather than generators." arXiv preprint arXiv:2003.10555 (2020).

[3] He, Pengcheng, et al. "Deberta: Decoding-enhanced bert with disentangled attention." arXiv preprint arXiv:2006.03654 (2020).

**Limitations:**

The authors discussed the limitations of this work in Section 7. However, the authors did not discuss the potential negative social impacts of this work. I look forward to discussing these issues with the authors and update my score accordingly.

**Strengths And Weaknesses:**

Strengths:
- proposes to leverage social media discourse for the task
- extensive experiments on 4 tasks, 10 datasets

Weaknesses:
- certain important experiment details are missing
- justification of the Hyperbolic Fourier Co-Attention for social text classification
- no societal impact discussions

---

> ### Author Response · Authors · 2022-08-01
> **Response to Reviewer niN4: Experimental details and justification of Hyperbolic Fourier co-Attention**
>
> We sincerely thank the reviewer for the valuable assessment and constructive feedback. We hereby address the raised questions and suggestions.
>
> ### Experimental details.
> > 1. It would be helpful for readers to know if the authors followed the splits of any previous work or created splits in these experiments.
>
> We followed a `train/test/val` split of `80:10:10` accross all 10 datasets.
>
> > 2. Did the authors implement "SOTA-1", "SOTA-2", and "SOTA-3" in Table 2 or directly quoted the reported results in these papers?
>
> The results of earlier works were quoted for the datasets with similar statistics and `train/test/val` splits (RumourEval, FigLang Reddit, and FigLang Twitter). For the rest of the datasets (Politifact, Gossipcop, Antivax, HASOC, Twitter15, Twitter16, and Pheme), all the data-specific and general baselines were retrained with the split `80:10:10`. Most of the used datasets were augmented and preprocessed to match our use case (**Section 5: *Datasets* and Appendix B: *Dataset preparation***), and as a result, the statistics (**Table 1**) varied as compared to their open-sourced versions used in previous works. **Appendix B: *Dataset Preparation*** discusses the details of dataset augmentation. We will clarify this in the final draft.
>
> > certain important experiment details are missing
>
> In addition to the above discussed details, experimental details have been laid out in **Section 5: *Experimentation details* and Appendix B: *Experimentation details***. We would be happy to shed more light on the same if the reviewer feels that some other details are missing, and add the same in the final draft.
>
>
>
> ### Justification of Hyperbolic Fourier Co-Attention for social text classification.
> > 3. I hope the authors could better justify how introducing the maths of Hyperbolic Fourier Co-Attention actually help Hyphen achieve this goal.
>
> - Yes, we agree that the notion of using social discourse is well-known. We hereby shed some intuition on the effectiveness of `Hyphen` in capturing the latent signals in public discourse. Often, the *public wisdom* expressed through the comments/replies to a social-text acts as a surrogate of crowd-sourced view and may provide us with complementary signals. However, these signals cannot always be readily used as in previous works.
> - To fuse the source post with the public discourse, the Fourier co-attention mechanism on the hyperbolic space computes pair-wise attention between user comments and the source post, thereby capturing the correlation between them. To capture this correlation, some existing studies like `dEFEND` compute vanilla co-attention between the two, or others like `TCNN-URG` simply concatenate them. This affects the interpretability of these models, resulting in *noise* seeping into the model, and thus performance degradation.
> - On a typical social media post, several users express their opinions, and several messages are being conveyed by the source post itself, some of which are more relevant and/or common than the others. Existing methods fail to capture these with their relative importance. The Fourier transform-based sublayer filters out the most-common user opinions expressed in the macro-AMR and the most prominent messages being conveyed by the source post. This also helps in reducing the noise in public discourse and filters out the most prominent social media *frequencies* like user opinions. We have elaborated more upon the intuition and usefulness of `Hyphen` in **Section 1: *Introduction***.

---

> ### Author Response · Authors · 2022-08-01
> **Response to Reviewer niN4: Comparision with newer LM's.**
>
> > 4. In the spirit of [1], I suggest the authors check out newer LMs [2,3] as sota baselines against Hyphen.
>
> Following the suggestion given by the reviewer to try out newer LMs, we have now compared the performance of `Hyphen` with `DeBERTa`[1], `Electra`[2], `DistilBERT`[3], and `AlBERT`[4]. Similar to BERT and RoBERTa, these models were fine-tuned on the source post content (without public discourse). The results are laid out as follows:
>
> | **Dataset** | **Metric** | **BERT** | **RoBERTa** | **DistilBERT** | **AlBERT** | **DeBERTa** | **Electra** | **Hyphen(E)** | **Hyphen(H)** |
> |:---:|---|:---:|:---:|:---:|:---:|:---:|:---:|:---:|:---:|
> | **Politifact** | Prec. | 0.911 | 0.924 | 0.912 | 0.911 | 0.937 | 0.906 | 0.951 | 0.972 |
> |  | Rec. | 0.904 | 0.903 | 0.911 | 0.904 | 0.932 | 0.904 | 0.936 | 0.961 |
> |  | F1 | 0.905 | 0.906 | 0.901 | 0.905 | 0.932 | 0.903 | 0.941 | 0.968 |
> | **Gossipcop** | Prec. | 0.764 | 0.771 | 0.761 | 0.711 | 0.782 | 0.746 | 0.786 | 0.791 |
> |  | Rec. | 0.761 | 0.775 | 0.765 | 0.717 | 0.785 | 0.751 | 0.776 | 0.788 |
> |  | F1 | 0.762 | 0.772 | 0.779 | 0.712 | 0.781 | 0.739 | 0.781 | 0.816 |
> | **ANTiVax** | Prec. | 0.943 | 0.948 | 0.927 | 0.934 | 0.926 | 0.922 | 0.941 | 0.951 |
> |  | Rec. | 0.941 | 0.961 | 0.935 | 0.932 | 0.915 | 0.952 | 0.937 | 0.927 |
> |  | F1 | 0.942 | 0.939 | 0.925 | 0.933 | 0.943 | 0.921 | 0.937 | 0.945 |
> | **HASOC** | Prec. | 0.646 | 0.647 | 0.632 | 0.688 | 0.651 | 0.675 | 0.712 | 0.748 |
> |  | Rec. | 0.651 | 0.661 | 0.621 | 0.621 | 0.636 | 0.636 | 0.703 | 0.718 |
> |  | F1 | 0.641 | 0.648 | 0.623 | 0.533 | 0.657 | 0.573 | 0.702 | 0.713 |
> | **Pheme** | Prec. | 0.861 | 0.852 | 0.853 | 0.855 | 0.871 | 0.857 | 0.854 | 0.877 |
> |  | Rec. | 0.862 | 0.851 | 0.851 | 0.848 | 0.865 | 0.858 | 0.843 | 0.875 |
> |  | F1 | 0.861 | 0.852 | 0.852 | 0.849 | 0.866 | 0.859 | 0.844 | 0.875 |
> | **Twitter15** | Prec. | 0.899 | 0.913 | 0.899 | 0.847 | 0.913 | 0.795 | 0.943 | 0.961 |
> |  | Rec. | 0.891 | 0.909 | 0.891 | 0.836 | 0.921 | 0.781 | 0.937 | 0.968 |
> |  | F1 | 0.891 | 0.908 | 0.894 | 0.834 | 0.929 | 0.778 | 0.936 | 0.957 |
> | **Twitter16** | Prec. | 0.921 | 0.895 | 0.881 | 0.901 | 0.919 | 0.931 | 0.944 | 0.946 |
> |  | Rec. | 0.918 | 0.891 | 0.897 | 0.897 | 0.918 | 0.861 | 0.936 | 0.937 |
> |  | F1 | 0.919 | 0.892 | 0.889 | 0.907 | 0.918 | 0.899 | 0.937 | 0.938 |
> | **RumourEval** | Prec. | 0.556 | 0.602 | 0.591 | 0.553 | 0.602 | 0.512 | 0.746 | 0.721 |
> |  | Rec. | 0.533 | 0.602 | 0.577 | 0.555 | 0.599 | 0.531 | 0.686 | 0.718 |
> |  | F1 | 0.533 | 0.595 | 0.571 | 0.549 | 0.612 | 0.565 | 0.697 | 0.712 |
> | **FigLang Twitter** | Prec. | 0.797 | 0.822 | 0.796 | 0.788 | 0.817 | 0.807 | 0.811 | 0.823 |
> |  | Rec. | 0.798 | 0.796 | 0.796 | 0.779 | 0.812 | 0.798 | 0.802 | 0.832 |
> |  | F1 | 0.797 | 0.801 | 0.795 | 0.784 | 0.811 | 0.808 | 0.812 | 0.822 |
> | **FigLang Reddit** | Prec. | 0.723 | 0.691 | 0.684 | 0.691 | 0.691 | 0.695 | 0.707 | 0.712 |
> |  | Rec. | 0.696 | 0.688 | 0.684 | 0.689 | 0.693 | 0.691 | 0.697 | 0.704 |
> |  | F1 | 0.677 | 0.689 | 0.683 | 0.693 | 0.692 | 0.682 | 0.698 | 0.701 |
>
> From the above results, it can be observed that `DeBERTa` consistently performs better than other transformer-based pre-trained models. `Electra`, `DistilBERT`, and `AlBERT` show comparable performance as `BERT` and `RoBERTa`. However, `Hyphen` outperforms all models (including `DeBERTa`) across all the tasks. This shows the effectiveness of `Hyphen` compared to SOTA language models. We will add these results in the final draft.
>
> [1] He, Pengcheng, et al. "Deberta: Decoding-enhanced bert with disentangled attention." arXiv preprint arXiv:2006.03654 (2020).
>
> [2] Clark, Kevin, et al. "Electra: Pre-training text encoders as discriminators rather than generators." arXiv preprint arXiv:2003.10555 (2020).
>
> [3] Sanh, Victor, et al. "DistilBERT, a distilled version of BERT: smaller, faster, cheaper and lighter." arXiv preprint arXiv:1910.01108 (2019).
>
> [4] Lan, Zhenzhong, et al. "Albert: A lite bert for self-supervised learning of language representations." arXiv preprint arXiv:1909.11942 (2019).

---

> ### Author Response · Authors · 2022-08-01
> **Response to Reviewer niN4: Annotation details and potential negative societal impacts.**
>
>
> ### Annotation details.
> > 5. I appreciate the authors' efforts in annotating the PolitiFact dataset for experiments in Section 6. Did the authors use crowdsourcing for these annotations? If so, please answer the questions in Checklist 5 accordingly.
>
> We sincerely apologize for any confusion from our end. We do not employ crowdsourcing for the annotation process but take the help of four expert annotators aged 25-30. The majority voting amongst the four annotators decided the final labels for a sentence. **Appendix C: *Data Annotation*** discusses these details about the annotation process and the annotators.
>
> ### Potential negative societal impacts.
>
> > **Limitations**: The authors discussed the limitations of this work in Section 7. However, the authors did not discuss the potential negative social impacts of this work. I look forward to discussing these issues with the authors and update my score accordingly.
> >
> We hereby discuss some potential negative societal impacts of this work, mitigation strategies, and how the existing `Hyphen` framework tackles them:
>
> **Malicious or unintended uses.** Our work does not involve a generative framework and hence cannot be used *directly* to generate DeepFakes. However, similar to other classification models like `dEFEND`, when combined with a generative model like a variational autoencoder, it could be misused to generate fake user comments. This could be mitigated with the help of our Fourier co-attention block. `Hyphen` adopts a Fourier transform-based network, which converts the comments' space to the *frequencies* domain. Thus, it cannot be used directly for generating fake user comments on social media posts because the user comment representations are now in a different semantic space. To do this, one would have to modify the source code and get rid of the Fourier block.
>
> **Environmental impact.** `Hyphen` architecture could be used to train large-scale models for various social-text classification tasks. However, owing to the numerical instabilities, the Riemannian optimization for Hyperbolic geometry is highly complex (e.g., Due to limited machine precision, the exponential and logarithmic maps might sometimes return points that are not exactly located on the manifold. To avoid this and to ensure that points remain on the manifold and tangent vectors remain on the right tangent space, we clamp the maximum norm). Thus, even though it is possible to train `Hyphen` on a large scale, it would not come without complications.
>
> **Fairness considerations.** The intuition behind `Hyphen` lies in finding the most prominent *frequencies* from the public discourse. However, often, the fake news (or hate speech) spreaders bias the comment section of a social media post to lure others into believing them. This could involve a fake news author creating fake accounts to comment in favor of his agenda and asking his connections to do the same. If the comment section is not credible, finding the majority voice of the people might bias the system in favor of the majority (which is a "pseudo-majority" as malicious users created it to fool the detection bots). To mitigate this, an additional module that estimates the credibility of user comments (and users) could assist the model in making fair decisions.
>
> **Privacy considerations.** Our work operates within the bounds of privacy considerations. We utilize the public discourse from social media to act as a surrogate for user opinions and worldly wisdom. However, we do not consider the user profile features (such as user background) as these might violate some privacy concerns and lead to ethical problems like "user profiling" [1]. Instead, we consider the user interpretation (opinions and wisdom) just w.r.t. the social-media post in hand.
>
> We will add this discussion on the potential negative societal impacts in the final draft.
>
> [1] Mann, Monique, and Tobias Matzner. "Challenging algorithmic profiling: The limits of data protection and anti-discrimination in responding to emergent discrimination." Big Data & Society 6.2 (2019): 2053951719895805.

---

> ### Author Response · Authors · 2022-08-08
> **Following up rebuttal response**
>
> We sincerely thank the reviewer for the valuable assessment and constructive feedback. We have tried to address the raised questions and suggestions in our rebuttal. We have carried out additional experimentation as suggested and have included a discussion on the potential negative impacts. Further, we have also elaborated on the annotation details and the justification of Hyperbolic Fourier co-Attention. Could you kindly let us know your feedback on the same?

---

> > ### Comment · Reviewer_niN4 · 2022-08-09
> > **Thank you for your response.**
> >
> > Thank you for your detailed response. My concerns are adequately addressed and I have updated my score to reflect that. I am especially impressed by the authors providing results for many PLMs on 10 datasets. You do have mighty computational resources at your lab. Also, sorry for not responding earlier since I was on vacation and just got back.

---

### Meta-Review · Area_Chair_s62m · 2022-08-25

**Recommendation:** Accept
**Confidence:** Certain

**Metareview:**

This paper proposed a discourse-aware hyperbolic spectral co-attention network for social text classification, via using public discourse and its hierarchy. Reviewers all agreed that this work presents an extensive amount of experiments/evaluations, with impressive performance gains and interpretability.  Thus, we recommend acceptance.


**Award:**

No

---

### Decision · Program_Chairs · 2022-09-14

Accept